# Effects of Land Use on the Mineralization of Organic Matter in Ultisol

Peidong Xu [2,3], Shihao Ma [2], Xiongfei Rao [1], Shipeng Liao [2], Jun Zhu [2] and Chunlei Yang [1,*]

1   Hubei Academy of Tobacco Sciences, Wuhan 430030, China
2   College of Resources and Environment, Huazhong Agricultural University, Wuhan 430070, China
3   Shanxi Key Laboratory of Grassland Ecological Protection and Native Grass Germplasm Innovation, College of Grassland Science, Shanxi Agricultural University, Taigu 030801, China
*   Correspondence: ycl193737@163.com; Tel.: +86-027-83606073

**Abstract:** Soil organic matter mineralization changed by land-use types is still not clearly understood. In this study, soils from typical land-use types including adjacent plantations of bamboo (Bam), camphor (Cam), and tea (Tea) were chosen to systematically investigate the role of organic carbon components and microbial community compositions in the organic matter mineralization in Ultisol. The mineralization of organic matter followed the sequence Bam < Cam < Tea. The higher carbon contents of labile pools were in the Cam and the Tea than that in the Bam. The carbon content of dissolved organic matter (DOM) showed the order Bam < Cam < Tea, whereas the complexity of chemical structure in DOM followed the opposite trend. The land-use types significant shifted the bacterial and fungal communities, and the relative abundances of bacterial or fungal phyla of *Actinobacteria*, *Acidobacteria*, *Firmicutes*, and *Basidiomycota* were significantly different among the land-use types. The multivariate regression tree results showed that the total organic carbon and/or the C/N ratio were dominant factors in influencing the bacterial and fungal communities. Moreover, the redundancy analysis results demonstrated that the communities of bacteria and fungi in Bam, Cam, and Tea were tightly linked to the C/N ratio, the pH and the labile pool I carbon, and the DOM, respectively. The Pearson's correlation results revealed that the mineralization of organic matter was significantly correlated with the organic carbon components, but generally not the microbial community compositions, which implied that the organic carbon components were perhaps the major determinant in controlling the organic matter mineralization in Ultisol.

**Keywords:** land-use types; mineralization; organic carbon components; microbial community compositions; Ultisol





## 1. Introduction

Soil organic matter mineralization is closely correlated with carbon dynamics, nutrient cycling, and climate change. Land-use types, such as plant species and associated with management practices, have generally changed the soil organic matter mineralization [1–5]. For example, the poplar plantations decreased the soil organic matter mineralization compared with the croplands [3]. Lin et al. demonstrated that soil organic matter mineralization was reduced when the land use was converted from broadleaf forests to moso bamboo plantations [4]. Therefore, it is very important to clearly understand the influence of land-use types on the soil organic matter mineralization.

Soil organic matter mineralization changed by the land-use types has generally been ascribed to the variations of organic carbon components and microbial community compositions [1–5]. For instance, the reduced soil organic matter mineralization rate with the conversion of broadleaf forests to bamboo plantations was closely linked to the decreased soil labile organic carbon fractions such as water-soluble organic carbon, readily oxidizable organic carbon, and O-alkyl carbon [4]. Zheng et al. observed that the soil organic matter mineralization rate in the croplands, higher than the poplar plantations, was positively

related with the bacterial abundance and the relative abundances of *Actinobacteria* and *Bacteroidetes*, while negatively correlated to the fungal abundance [3]. However, the majority of these studies only emphasized the role of organic carbon components or microbial community compositions in the soil organic matter mineralization.

Organic carbon components play various roles in the soil organic matter mineralization. Labile pools of soil organic carbon were shown to be the easily degradable substrate compared with the recalcitrant pool [6–8]. Dissolved organic matter (DOM), especially with a simple chemical structure, can facilitate the soil organic matter mineralization [9–11]. The breakdown of soil organic matter is restricted at the high C/N ratio [1,12].

The microbial community compositions have multiple influences on the soil matter mineralization. Fast-growing copiotrophic bacteria such as *Bacteroidetes* and *Proteobacteria* were reported to have higher metabolic capacities for decomposing soil organic matter than that of slow-growing oligotrophic bacteria such as *Acidobacteria* and *Chloroflexi* [3,5,7,8,13,14]. In the fungal community, *Ascomycota* can degrade cellulose, while *Basidiomycota* can play critical role in cellulose and lignin decomposition [15–17]. Moreover, Zygomycetes can utilize the easily acquired sugars from the fresh substrate [18].

Ultisol is widely distributed across subtropical and tropical zones of southern China. Ultisol suffers from strong weathering and leaching and has unique characteristics such as high contents of clay and Fe/Al and low contents of organic matter and nutrients [7,10]. Therefore, it is very important to clearly understand the roles of both organic carbon components and microbial community compositions in the organic matter mineralization of Ultisol.

In this study, the soils from typical land-use types including adjacent single-species plantations of bamboo, camphor, and tea were chosen to systematically investigate the effects of both organic carbon components and microbial community compositions on the organic matter mineralization in the Ultisol. We hypothesized that (i) the organic matter mineralization, the organic carbon components, and the microbial community compositions were different among the land-use types; (ii) the land-use types changed the soil organic matter mineralization mainly by the organic carbon components rather than the microbial community compositions.

## 2. Materials and Methods

### 2.1. Site Description and Soil Sample Collection

The research site was located in Xianning County (30°01′ N, 114°21′ E), Hubei province, China. The study area is categorized as having a subtropical climate with a mean annual temperature of 16.8 °C and a mean annual precipitation of 1300 mm. The soil type is Ultisol derived from Quaternary red clay [19]. In this area, the adjacent single-species plantations of bamboo (Bam, *Bambusoideae*), camphor (Cam, *Cinnamomum camphora*), and tea (Tea, *Camellia sinensis*) selected for this study were the typical land-use types over 50 years. The bamboo plantation has been moderately disturbed because of the occasional harvesting of bamboo stems and shoots. The camphor plantation is in the nearly natural state and has experienced hardly any human disturbance throughout the period. A total of 1500 kg ha$^{-1}$ compound fertilizer (N-P$_2$O$_5$-K$_2$O: 20%-8%-8%) has been applied to the tea plantation and the branches are pruned and deposited as a surface mulch every year.

Three independent plots (20 m × 20 m) of each land-use type were randomly selected for soil sampling. The soils from ten randomly selected points (about 30 cm from the plant stand) were collected from the 0–20 cm layer by 5 cm diameter auger and mixed homogenously to form a composite sample in each plot. The soil samples were placed in sterile plastic bags and immediately transported to the laboratory in an iced cooler box. Visible roots and rocks were removed, and the soils were sieved through 2 mm to prepare into three parts. The first part was air-dried to measure soil basic properties and organic carbon components. The second part was stored at 4 °C to determine organic matter mineralization and enzyme activities. The last part was stored at −80 °C for microbial community analysis.

### 2.2. Basic Properties of Soils

Total organic carbon (TOC) and total nitrogen (TN) contents were determined using the methods of Walkley–Black and Kjeldahl [20,21], respectively. In brief, the content of TOC was measured by potassium dichromate oxidation followed by titration with ferrous sulfate. Moreover, the content of TN was determined by the continuous steps of sulfuric acid digestion, sodium hydroxide distillation, boric acid uptake, and diluted acid titration. Soil pH was measured by using a digital pH meter with soil/water ratio of 1:2.5 (*w/v*).

### 2.3. Mineralization of Soil Organic Matter

The mineralization of soil organic matter followed the method of Xu et al. [7]. Briefly, 150 g of dry-equivalent soils was moistened to 60% of the water-holding capacity and aerobically incubated at 25 °C in the dark for 121 days in 1250 mL flasks. Before the incubation, the soils were pre-incubated under the same conditions for 5 days to stabilize the microbial activity [2]. The soils were maintained at a constant moisture by adding sterile distilled water in the incubation period. The caps of flasks were opened to replenish oxygen with the exception of regular $CO_2$ collection. The $CO_2$ released from soils was collected by syringes from the headspace of flasks and was measured on 1, 2, 4, 7, 11, 16, 22, 29, 37, 46, 56, 86, 101, and 121 days by a gas chromatograph (Agilent, G7890A, Wilmington, DE, USA). The $CO_2$ released from soils was regularly collected from the headspace of flasks and was measured using a gas chromatograph (Agilent, G7890A, Wilmington, DE, USA). The single exponential model was used to fit the cumulative curves of $CO_2$-C release over time [22]:

$$C_t = C_0 \left[ 1 - \exp \left( -k \times t \right) \right]$$

where $C_t$ (mg $CO_2$-C $kg^{-1}$) is the cumulative amount of mineralized organic carbon at the time of t (d), $C_0$ (mg $CO_2$-C $kg^{-1}$) is the size of potentially mineralizable organic carbon pool, and k ($d^{-1}$) is the mineralization rate constant of organic carbon.

### 2.4. Organic Carbon Pools and Dissolved Organic Matter of Soils

Soil organic carbon pools were classified by the two-step acid hydrolysis method [6]. Briefly, 0.5 g of soils was hydrolyzed with 20 mL of 2.5 M $H_2SO_4$ and maintained at 105 °C for 30 min. The organic carbon in hydrolysate was defined as the labile pool I carbon (LP I-C). The remaining residue was continued to hydrolyze with 2 mL of 13 M $H_2SO_4$ at room temperature overnight, and then the acid was diluted to 1 M and maintained at 105 °C for 3 h. The organic carbon in this hydrolysate was recognized as the labile pool II carbon (LP II-C). The above hydrolysates were obtained by centrifuging at 10,000 rpm for 30 min and then filtering through a 0.45-μm membrane filter. The organic carbon in the hydrolysates was determined by a TOC analyser (Vario TOC, Elementar, Germany). The organic carbon in the last residue was determined as the recalcitrant pool carbon (RP-C), calculated as the difference between the total organic carbon and the labile pool carbon.

Soil dissolved organic matter (DOM) was extracted by deionized water with a solid to water ratio of 1:5 (*w/v*) [10]. The carbon concentration of DOM was measured by a TOC analyser (Vario TOC, Elementar, Germany). The chemical structure of DOM was evaluated by using a fluorescence spectrophotometer (Jasco, FP-6500, Tokyo, Japan) [10,11]. In order to normalize fluorescence intensity (FI), all of the DOMs were adjusted by deionized water to the same carbon concentration before the spectroscopic measurement. The fluorescence excitation–emission matrices (EEMs) of DOM were recorded over the emission wavelength range of 300–550 nm at an excitation wavelength of 250–450 nm with a 5 nm increment. The structural difference of DOM among the land-use types was inferred by the location and/or normalized FI of the fluorophore in EEMs [23,24].

### 2.5. Microbial Communities and Enzyme Activities of Soils

Soil microbial communities were determined by high-throughput sequencing. The DNA of soils was extracted by a FastDNA Spin Kit for Soil (Mpbio, Carlsbad, CA, USA). The quantity and quality of final DNA were checked using a Nanodrop-2000 spectropho-

tometer (NanoDrop Technologies, Wilmington, DE, USA). The bacterial 16S rRNA and fungal ITS genes were amplified by using primers of 338F-806R and ITS1F-2043R, respectively [14]. The purified amplicons were pooled in equimolar and sequenced on an Illumina MiSeq PE300 platform (Illumina Inc.; San Diego, CA, USA). The generated raw sequences were quality-filtered by Trimmomatic and merged together by FLASH. The operational taxonomic units (OTUs) were clustered with a 97% similarity cutoff by using UPARSE. The observed OTUs, Chao 1, and Shannon indexes were calculated to describe the diversity of the microbial community. The shifts in microbial communities were visualized by the principal co-ordinates analysis (PCoA) ordination plot based on the Bray–Curtis distances. The bacterial and fungal taxonomy were assigned by using an RDP classifier against the SILVA and UNITE databases, respectively. The activities of cellulase and phenol oxidase were assayed as described by Guan [25].

*2.6. Statistical Analysis*

A one-way ANOVA was conducted followed by the Duncan test to analyze significant differences ($p < 0.05$) among the land-use types. A permutational multivariate analysis of variance (PERMANOVA) was implemented to estimate the effects of land-use types on the bacterial and the fungal communities. Multivariate regression tree (MRT) analysis was performed to determine the dominant environment factors in influencing the bacterial and fungal communities. Redundancy analysis (RDA) was applied to assess the relationships between the environment factors and the bacterial and the fungal communities. Pearson's correlation analysis was carried out to evaluate the relationship between the organic matter mineralization and the organic carbon components as well as the microbial community compositions. These analyses were employed in R 4.0 software associated with vegan and mvpart packages.

**3. Results**

*3.1. Basic Properties of Soils under the Land-Use Types*

The contents of total organic carbon (TOC) in Bam, Cam, and Tea soils were 9.67, 12.70, and 13.29 g kg$^{-1}$, respectively (Table 1). The total nitrogen (TN) contents in Bam, Cam, and Tea soils were 0.94, 1.42, and 1.45 g kg$^{-1}$, respectively (Table 1). Both the contents of TOC and TN were significantly different among the land-use types (Table 1). Moreover, the ratio of carbon to nitrogen (C/N) followed the order Cam < Tea < Bam, and the difference was significant (Table 1). The Cam had the highest soil pH, while the lowest soil pH was observed in the Tea (Table 1).

**Table 1.** Basic properties of soils from bamboo (Bam), camphor (Cam), and tea (Tea) plantations.

|  | TOC (g kg$^{-1}$) | TN (g kg$^{-1}$) | C/N | pH |
|---|---|---|---|---|
| Bam | 9.67 c | 0.94 c | 10.26 a | 4.55 b |
| Cam | 12.70 b | 1.42 b | 8.93 c | 4.85 a |
| Tea | 13.29 a | 1.45 a | 9.16 b | 4.29 c |

TOC, total organic carbon; TN, total nitrogen; C/N, ratio of carbon to nitrogen. Values with the same letters within a column were not statistically different at $p < 0.05$.

*3.2. Mineralization of Soil Organic Matter under the Land-Use Types*

The cumulative amount of soil organic matter mineralization within 121 days of incubation was significantly different among the land-use types, and the cumulative $CO_2$-C released from Bam, Cam, and Tea soils was 523.9, 953.6, and 1015.9 mg kg$^{-1}$, respectively (Figure 1b). Moreover, the size ($C_0$) and proportion ($C_0$/TOC) of potentially mineralizable organic carbon pool showed the sequence Bam < Cam < Tea (Table 2).

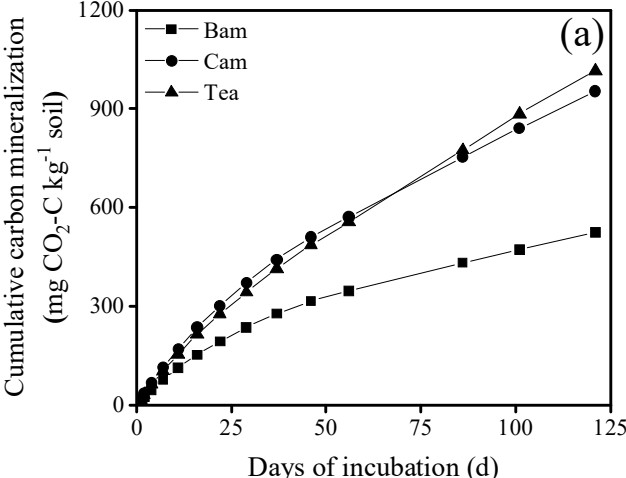
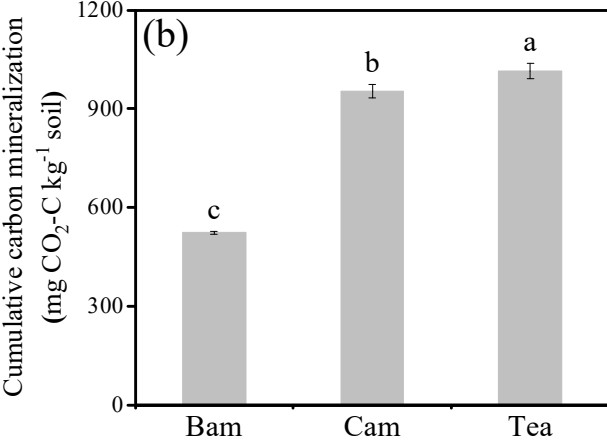

**Figure 1.** Cumulative amount of soil organic matter mineralization over incubation days (**a**) and within 121 days of incubation (**b**) in bamboo (Bam), camphor (Cam), and tea (Tea) plantations. The different letters adjacent to the bars were statistically different at $p < 0.05$.

**Table 2.** Mineralization parameters of soil organic matter in bamboo (Bam), camphor (Cam), and tea (Tea) plantations.

| | Fitting Parameters | | | $C_0$/TOC (%) |
|---|---|---|---|---|
| | $C_0$ (mg CO$_2$-C kg$^{-1}$) | k (d$^{-1}$) | $R^2$ | |
| Bam | 568.5 | 0.0180 | 0.996 | 5.88 |
| Cam | 1181.3 | 0.0126 | 0.997 | 9.30 |
| Tea | 1595.3 | 0.0081 | 0.997 | 12.00 |

$C_0$, potentially mineralizable organic carbon pool; k, mineralization rate constant of organic carbon.

### 3.3. Organic Carbon Pools and Dissolved Organic Matter of Soils under the Land-Use Types

The organic carbon content of the labile pool I (LP I-C) in Bam, Cam, and Tea soils was 3.43, 4.35, and 4.15 g kg$^{-1}$, respectively (Table 3). The labile pool II carbon (LP II-C) content in Bam, Cam, and Tea soils was 1.83, 2.37, and 2.34 g kg$^{-1}$, respectively (Table 3). The content of recalcitrant pool (RP-C) followed the order Bam < Cam < Tea (Table 3). All of the contents of organic carbon pools were significantly different among the land-use types (Table 3).

**Table 3.** Labile and recalcitrant pools of soil organic carbon, and soil dissolved organic matter in bamboo (Bam), camphor (Cam), and tea (Tea) plantations.

| | LP I-C (g kg$^{-1}$) | LP II-C (g kg$^{-1}$) | RP-C (g kg$^{-1}$) | DOM (mg C kg$^{-1}$) |
|---|---|---|---|---|
| Bam | 3.43 c | 1.83 b | 4.40 c | 92.99 b |
| Cam | 4.35 a | 2.37 a | 5.98 b | 96.48 b |
| Tea | 4.15 b | 2.34 a | 6.80 a | 123.79 a |

LP I-C, labile pool I of soil organic carbon; LP II-C, labile pool II of soil organic carbon; RP-C, recalcitrant pool of soil organic carbon; DOM, dissolved organic matter. Values with the same letters within a column were not statistically different at $p < 0.05$.

The content of soil DOM in Bam, Cam, and Tea was 92.99, 96.48, and 123.79 mg C kg$^{-1}$, respectively (Table 3). The fluorophore of DOM in Bam, Cam, and Tea soils was located at 370 ex/427 em, 355 ex/405 em. and 350 ex/402 em of excitation/emission wavelength pairs, respectively (Figure 2). These locations of fluorophore showed that the humic acid-like organics were dominant in DOM from each land-use type [26]. Moreover, the normalized fluorescence intensity (FI) of fluorophore followed the sequence Bam < Cam < Tea (Figure 2). The fluorophore at the long wavelength pair with low FI was probably due to the complex

structure of DOM with the high degree of conjugation and high molecular weight [10,23,24]. On the other hand, the fluorophore at the short wavelength pair with high FI was perhaps attributed to the simple structure of DOM with the low degree of polycondensation and low molecular weight [10,23,24]. Thus, the structural complexity of DOM was increased as the order Tea < Cam < Bam.

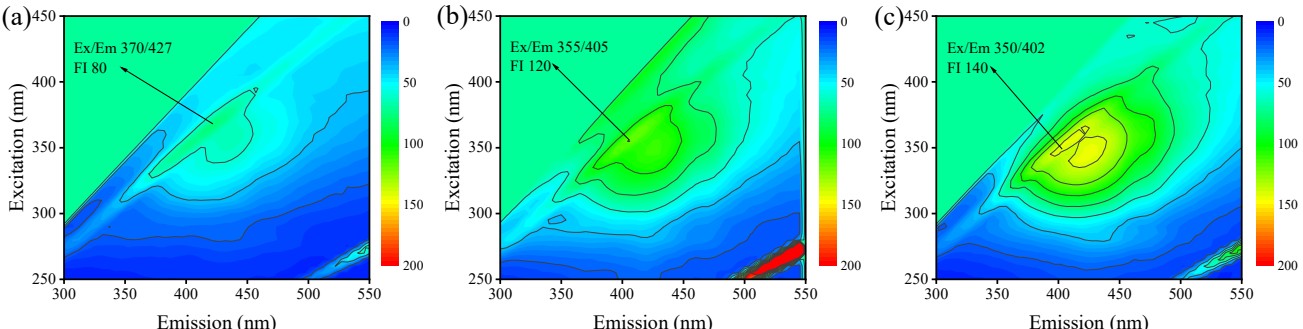

**Figure 2.** Fluorescence excitation–emission matrices (EEMs) of soil DOM from bamboo (**a**), camphor (**b**), and tea (**c**) plantations. Contour lines based on DOM-normalized fluorescence intensity (FI) values. Arrows showing the location and normalized FI value of the fluorophore.

*3.4. Microbial Communities and Enzyme Activities of Soils under the Land-Use Types*

The bacterial OTUs, Chao 1, and Shannon indexes in Cam soil were higher than that in Bam and Tea soils, and the difference was significant among the land-use types (Table 4). The PCoA plots and PERMANOVA results revealed the clear variations in bacterial community among the land-use types (Figure 3a). The dominant phyla of soil bacteria in the three land-use types included *Proteobacteria*, *Chloroflexi*, *Actinobacteria*, *Acidobacteria*, and *Firmicutes*, which accounted for 24.1–27.8%, 23.1–25.7%, 14.5–26.9%, 11.6–20.5%, and 2.9–9.3% of the bacterial sequences, respectively (Figure 4a). The slightly higher relative abundances of *Proteobacteria* and *Chloroflexi* were observed in the Cam soil than that in Bam and Tea soils (Figure 4c). The relative abundances of *Actinobacteria*, *Acidobacteria*, and *Firmicutes* were significantly different among the land-use types (Figure 4c). Moreover, the Bam, Cam, and Tea soils contained the highest relative abundance of *Actinobacteria*, *Acidobacteria*, and *Firmicutes*, respectively (Figure 4c).

**Table 4.** Soil bacterial and fungal diversities and enzyme activities in bamboo (Bam), camphor (Cam), and tea (Tea) plantations.

|  | Bacteria Diversity | | | Fungi Diversity | | | Cellulase | Phenol Oxidase |
|---|---|---|---|---|---|---|---|---|
|  | OTUs | Chao 1 | Shannon | OTUs | Chao 1 | Shannon | (mg Glucose $g^{-1}$ 72 $h^{-1}$) | (mg Purple Pyrogallol $g^{-1}$ 2 $h^{-1}$) |
| Bam | 1009 b | 1314 b | 5.08 c | 244 a | 292 a | 2.79 a | 0.23 b | 0.29 b |
| Cam | 1256 a | 1533 a | 5.99 a | 225 a | 299 a | 2.33 c | 0.38 a | 0.31 b |
| Tea | 1021 b | 1221 c | 5.73 b | 158 b | 192 b | 2.51 b | 0.42 a | 0.36 a |

Values with the same letters within a column were not statistically different at $p < 0.05$.

The Bam soil with higher OTUs, Chao 1, and Shannon indexes of fungal community than that in Cam and Tea soils, and the land-use types significantly changed the fungal diversity (Table 4). The PCoA plots and PERMANOVA results revealed the clear variations in fungal community among the land-use types (Figure 3b). The dominant phyla of soil fungi in the three land-use types were *Zygomycota*, *Ascomycota*, and *Basidiomycota*, which accounted for 40.3–57.6%, 20.7–24.1%, and 0.5–12.9% of the fungal sequences, respectively (Figure 4b). A slightly higher relative abundance of *Zygomycota* was found in Tea soils than that of Bam and Cam soils, while the Cam soil had a slightly higher relative abundance of *Ascomycota* than that of the Bam and Tea soils (Figure 4d). The Tea soil

contained significantly higher relative abundances of *Basidiomycota* than that of Bam and Cam soils (Figure 4d).

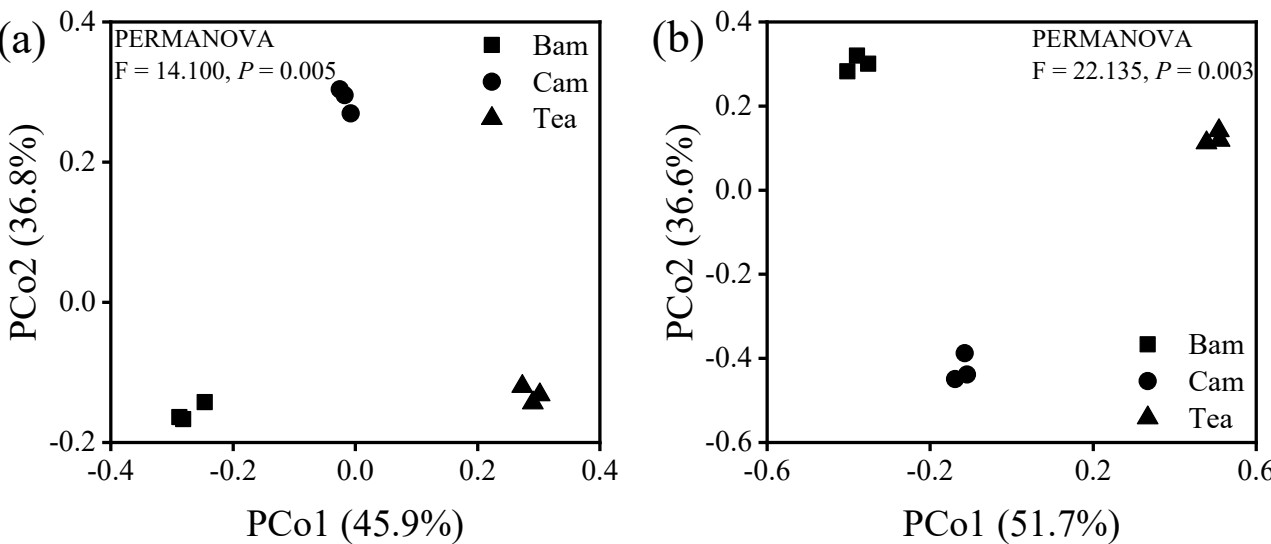

**Figure 3.** Principal co-ordinates analysis (PCoA) ordination plots of soil bacterial (**a**) and fungal (**b**) communities from bamboo (Bam), camphor (Cam), and tea (Tea) plantations.

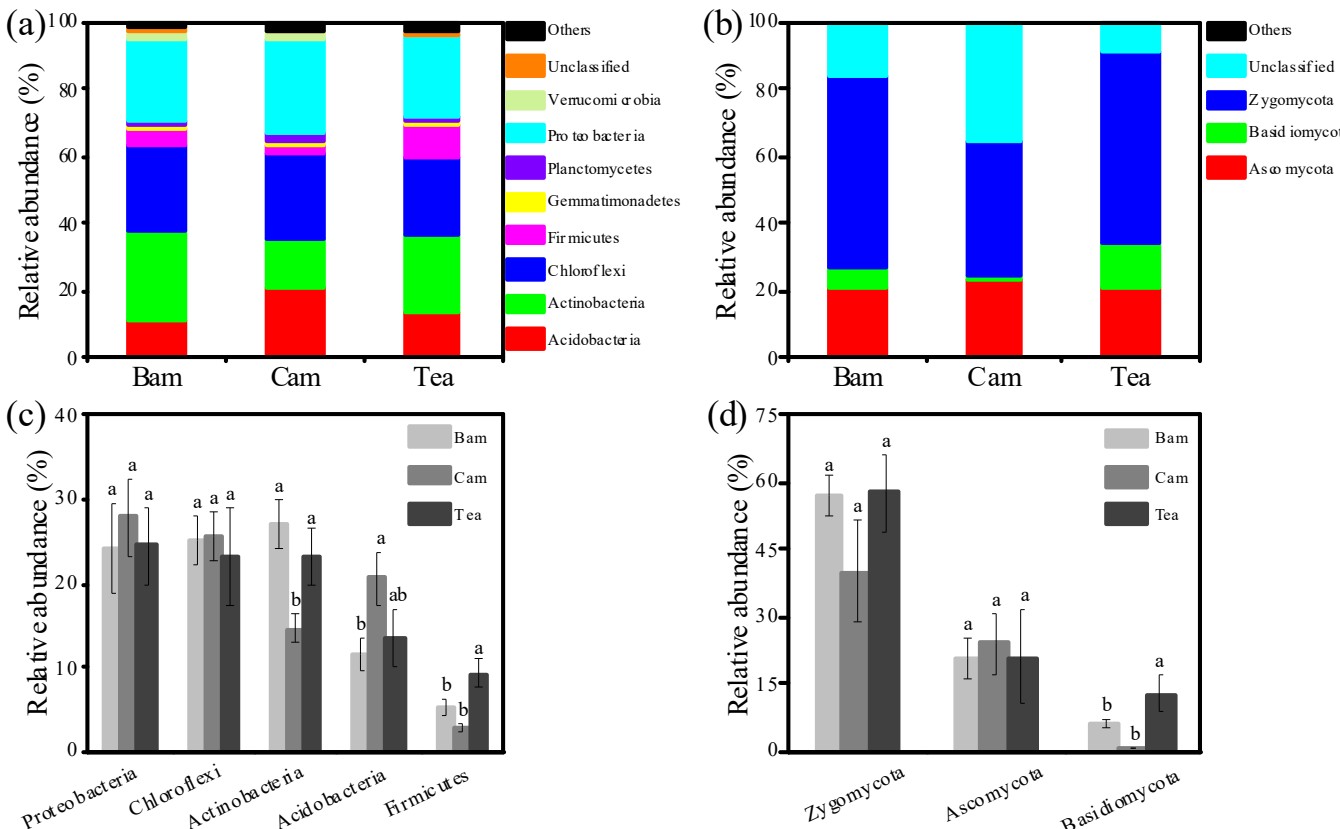

**Figure 4.** Relative abundances of soil bacterial (**a**) and fungal (**b**) phyla, and the dominant bacterial (**c**) and fungal (**d**) phyla in bamboo (Bam), camphor (Cam), and tea (Tea) plantations. The same letters were not statistically different at $p < 0.05$.

The activities of cellulase and phenol oxidase were significantly different and followed the sequence Bam < Cam < Tea (Table 3).

### 3.5. Relationships between Soil Properties, Organic Carbon Components, Microbial Community Compositions, and Mineralization

The MRT analysis explained 82.98% of the variation in the bacterial community (Figure 5a). The Bam was separated from the other treatments by the TOC content ($</\geq 11.2$ g kg$^{-1}$) in the first split, which interpreted 65.29% of the variance (Figure 5a). The TOC content ($</\geq 13.04$ g kg$^{-1}$) further separated the Cam and the Tea in the second split, which defined 17.69% of the variance (Figure 5a). The MRT analysis explained 83.02% of the variance in the fungal community (Figure 5b). The Bam associated with higher C/N ($\geq 9.68$) were separated from the other treatments in the first split, clarifying 44.36% of the variance (Figure 5b). The TOC content ($</\geq 13.04$ g kg$^{-1}$) in the second split then separated the Cam and the Tea, identifying 38.66% of the variance (Figure 5b).

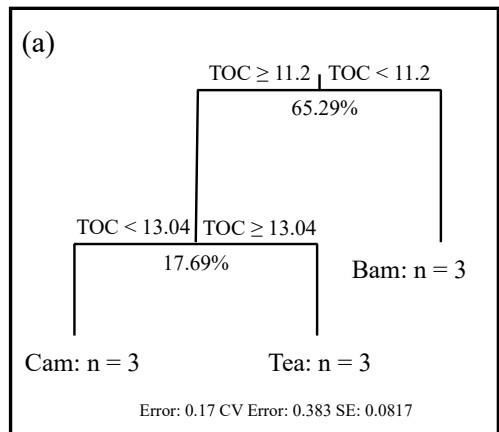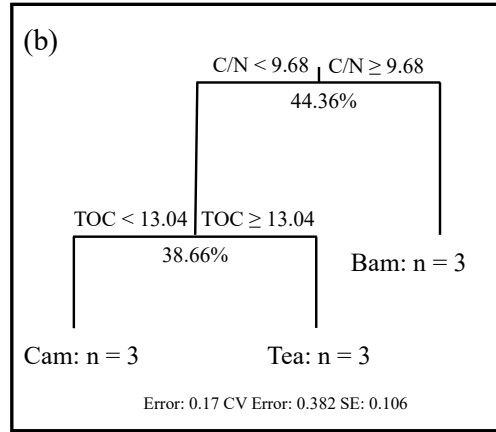

**Figure 5.** Multivariate regression tree (MRT) of soil bacterial (**a**) and fungal (**b**) communities from bamboo (Bam), camphor (Cam), and tea (Tea) plantations. TOC, total organic carbon; C/N, ratio of carbon to nitrogen.

The first two axes of the RDA accounted for 82.02% and 84.49% of the total variance in the bacterial and the fungal communities, respectively (Figure 6). Moreover, both the bacterial and the fungal communities from Bam, Cam, and Tea were positively correlated with the C/N ratio, the pH, and the LP I-C and the DOM, respectively (Figure 6).

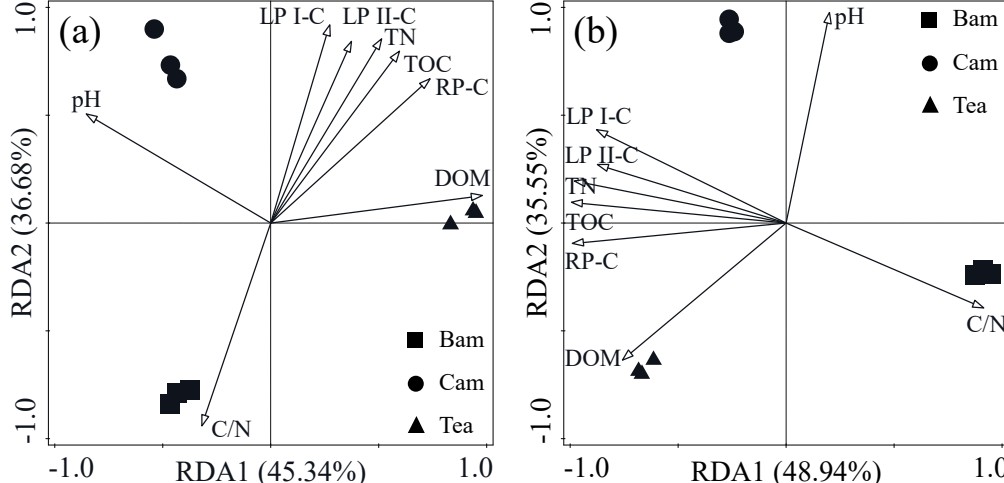

**Figure 6.** Redundancy analysis (RDA) of soil bacterial (**a**) and fungal (**b**) communities from bamboo (Bam), camphor (Cam), and tea (Tea) plantations. TOC, total organic carbon; TN, total nitrogen; C/N, ratio of carbon to nitrogen; LP I-C, labile pool I of soil organic carbon; LP II-C, labile pool II of soil organic carbon; RP-C, recalcitrant pool of soil organic carbon; DOM, dissolved organic matter.

The mineralization of organic matter showed significant relationships with the contents of TOC, TN, LP I-C, LP II-C, DOM, the C/N ratio, and the cellulase activity, but a nonsignificant correlation with the microbial community compositions with the exception of *Gemmatimonadetes* (Table 5).

**Table 5.** Pearson's correlation between soil organic matter mineralization and soil properties, organic carbon components as well as microbial community compositions. Significant difference at $p < 0.05$ is given in bold.

|  | r | p |
|---|---|---|
| TOC | 0.996 | **<0.001** |
| TN | 0.993 | **<0.001** |
| C/N | −0.951 | **<0.001** |
| pH | −0.073 | 0.852 |
| LP I-C | 0.935 | **<0.001** |
| LPII-C | 0.904 | **0.001** |
| DOM | 0.667 | **0.050** |
| Cellulase | 0.851 | **0.004** |
| Phenol oxidase | 0.641 | 0.063 |
| Acidobacteria | 0.438 | 0.238 |
| Actinobacteria | −0.558 | 0.119 |
| Chloroflexi | −0.141 | 0.718 |
| Firmicutes | 0.261 | 0.497 |
| Gemmatimonadetes | 0.742 | **0.027** |
| Planctomycetes | 0.037 | 0.925 |
| Proteobacteria | 0.192 | 0.621 |
| Verrucomicrobia | −0.644 | 0.061 |
| Ascomycota | 0.109 | 0.780 |
| Basidiomycota | 0.147 | 0.706 |
| Zygomycota | 0.223 | 0.564 |

TOC, total organic carbon; TN, total nitrogen; C/N, ratio of carbon to nitrogen; LP I-C, labile pool I of soil organic carbon; LP II-C, labile pool II of soil organic carbon; DOM, dissolved organic matter.

## 4. Discussion

### 4.1. Influences of the Land-Use Types on the Soil Organic Carbon Components

In the present study, the labile pools in soil organic carbon (LP I-C and LP II-C) followed the sequence Bam < Tea ≤ Cam, and the recalcitrant pool in soil organic carbon was in the order Bam < Cam < Tea (Table 3). Different from our results, Wang and Zhong found that the labile and recalcitrant pools of organic carbon in the surface soil were similar among the four monoculture forests [1]. However, Lin et al. noted that the broadleaf forest had more of the soil labile organic carbon fraction (O-alkyl C) compared with the bamboo plantations [4]. The dissimilarity among the above studies is possibly related to the variations in vegetation species, environmental conditions, and management practices [4,27].

LP I-C consists of polysaccharides such as hemicellulose and starch, and LP II-C predominantly contains cellulose [6,9]. The components of soil organic carbon are potentially influenced by the input of litter-fall and root from different plantations [28]. The broadleaf litter-fall with a low C/N ratio is easily decomposed and is probably an important source of labile fractions in the Cam soil [18,28]. The fertilization in the Tea perhaps stimulates the root activity and promotes the large amounts of labile exudates into the soil [10]. Moreover, the mulch of regular pruning in the tea plantation could contribute to the high soil organic matter. Ultisol belongs to strongly weathered soil and has a limited capacity for organic matter accumulation, especially in the more disturbed bamboo plantation.

DOM is the important and typical labile component in soil organic carbon. The Cam and the Tea showed more contents of DOM in soils than that in the Bam (Table 3), the variation was similar to the contents of LP I-C and LP II-C among the land-use types. Similarly, Lin et al. suggested that the contents of water-soluble organic carbon decreased

when broadleaf forests were converted to moso bamboo plantations in two soil layers [4]. High precipitation in this area could stimulate DOM leaching, particularly in the more disturbed bamboo plantation. Importantly, the complex of chemical structure of DOM was in the sequence Tea < Cam < Bam (Figure 2). The Cam and the Tea provided more plant materials, which would leach abundant DOMs with simple structures compared with the Bam, because the DOM from plant materials mostly showed low molecular complexity and humification indices [10]. Furthermore, the higher activities of enzymes (Table 4) perhaps resulted in more input of DOMs with simple structure in the Cam and the Tea than that in the Bam [11]. These results implied that the Cam and the Tea contained more, and simpler labile organic carbon components compared with the Bam.

### 4.2. Influences of the Land-Use Types on the Soil Microbial Communities

Our study revealed clear shifts in both bacterial and fungal communities among the land-use types (Figure 3). The MRT results indicated that the TOC content and/or the C/N ratio were the important factors in separating the microbial community among the land-use types (Figure 5). Indeed, the soil organic carbon components are the critical determinant of microbial communities [17,29], and microbial category can be classified into functional groups based on their response to organic carbon components and soil environment [13,30,31].

The RDA results further suggested that the microbial communities of Bam, Cam, and Tea were dominantly influenced by the C/N ratio, the LP I-C and the soil pH, and the DOM, respectively (Figure 6). The high relative abundance of *Actinobacteria* was in the Bam soil, which could be due to the high soil C/N ratio (Figures 4c and 6a). Urbanová et al. reported the high abundance of *Actinobacteria* in litters with high C/N ratios [32]. Moreover, *Actinobacteria* are able to utilize more recalcitrant organic matter by their hypha penetration [33]. The high relative abundance of *Proteobacteria*, *Chloroflexi*, *Acidobacteria*, and *Ascomycota* were from the Cam soil, which was perhaps attributed to the high LP I-C content and/or the high soil pH (Figure 4c,d and Figure 6). In fact, *Proteobacteria* (such as *Alpha-*, *Beta-*, and *Gamma-Proteobacteria*) exhibit copiotrophic attributes and are more abundant in soils with large amounts of available carbon [3,5,7,13,14]. *Acidobacteria* can show the diverse use of polysaccharides including hemicellulose, cellulose, xylan, and chitin [16,34]. Liu et al. and Li et al. observed that some *Acidobacteria* subgroups were more abundant in soils with a higher organic carbon content [29,35]. Additionally, Rousk et al. and Liu et al. reported that *Chloroflexi* phylum and some subgroups of *Proteobacteria* and *Acidobacteria* phyla tended to be positively related to soil pH [31,35]. *Ascomycota* perhaps predominate in cellulose-rich (primary component of labile pools) soils [16,17]. The high relative abundance of *Firmicutes*, *Zygomycota*, and *Basidomycota* were in the Tea soil, which was probably ascribed to the high DOM content (Figure 4c,d and Figure 6). Indeed, *Firmicutes* generally prefer a copiotrophic soil environment with easily acquired organic carbon [2,17]. Some species such as *Zygomycetes* in the *Zygomycota* phylum could be favored by the readily available sugars and adequate nutrients of fresh substrate [2,18]. Ren et al. showed that the restored ecosystem could promote the fast growth of *Basidomycota* by the utilization of an easily degradable fraction of residues [5].

### 4.3. Influences of the Land-Use Types on the Mineralization of Soil Organic Matter

Our study suggested that the mineralization of soil organic matter significantly increased in the sequence Bam < Cam < Tea (Figure 1b). The mineralization of soil organic matter showed a significantly positive relationship with the contents of TOC, TN, LP I-C, LP II-C and DOM, and the activity of cellulase, whereas it revealed a significantly negative relationship with the C/N ratio (Table 5). The labile organic carbon, especially the DOM with simple chemical structure, could provide small and available substrate and potentially promote the microbial metabolism and $CO_2$ production [2,5,6,9–11]. Soil with a high nitrogen content and a low C/N ratio tends to accelerate the degradation of organic matter [1,12]. Cellulase is primarily involved in the decomposition of cellulose in soil [36].

The soil organic polymers broken down by the enzymes can supply small, dissolved, and available organic components for microorganisms [7].

However, the mineralization of soil organic matter generally showed nonsignificant relationships with the microbial community compositions (Table 5). These results implied that organic carbon components played a more important role than microbial community compositions in the mineralization of organic matter influenced by the land-use types in the Ultisol. Indeed, the influence of microbes on the mineralization of soil organic matter generally depends on the organic substrate [2,7,8,10]. Therefore, the higher contents of carbon and nitrogen, the lower C/N ratio, and the higher cellulase activity in the Cam and the Tea largely stimulated the greater mineralization of soil organic matter compared with the Bam.

## 5. Conclusions

The soil organic matter mineralization showed the order Bam < Cam < Tea. The Cam and Tea contained higher contents of labile organic carbon and a simpler chemical structure of DOM than that in the Bam. The microbial communities were significantly different among the land-use types. The relative abundances of *Actinobacteria*, *Acidobacteria*, *Firmicutes*, and *Basidiomycota* exhibited significant difference among the land-use types. The microbial communities were dominantly controlled by the TOC content and/or the C/N ratio. The microbial communities of Bam, Cam, and Tea displayed close correlations with the C/N ratio, the pH and LP I-C, and the DOM, respectively. Land-use types changed the organic matter mineralization mainly by the organic carbon components rather than the microbial community compositions in the Ultisol.

**Author Contributions:** P.X.; S.M.; X.R.; S.L. and C.Y. supervised and designed the project; P.X. performed the experiment and collected data; P.X. and J.Z. helped in data formal analysis and preparation of the manuscript; J.Z. and C.Y. reviewed the manuscript. All authors have read and agreed to the published version of the manuscript.

**Funding:** This research was funded by the Science and Technology Project in Hubei Tobacco Company (Grant No. 027Y2021-009), the Basic Research Program of Shanxi Province (Grant No. 202103021223128), the Research projects of Shanxi Province's doctoral graduates and postdoctoral researchers working in Shanxi Province (Grant No. SXBYKY2022026), and the Science and Technology Innovation Fund of Shanxi Agricultural University (Grant No. 2021BQ62).

**Data Availability Statement:** Not applicable.

**Conflicts of Interest:** The authors declare no conflict of interest.

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
