# Peer review of "Effects of Land Use on the Mineralization of Organic Matter in Ultisol"

_agronomy, doi:10.3390/agronomy12122915_

Round 1
Reviewer 1 Report
Dear Author,
Please find below general comments and by-line suggestion of revisions.
General Comments
1. The introduction should contain all the necessary information to justify the measurements. For instance, it should be introduced concisely about the Ultisol, in terms of organic matter contains and its related soil properties (such as N, C/N ratio)
2. Sampling collection should explain in more detail information, i.e., i) tool used for soil sampling (like gauge auger), ii) the distance of soil sampling from the plant stand
3. This part needs more detailed information.
In the methods section of 2.4 sub section, the method of fractionation did not apply/ or less information of substantial part as the reference paper (Rovira and Vallejo, 2022), like process of centrifugation, shaking the solution and drying the residue. Also, the tool used to analyse C concentration of labile C (LP I C, LP II C) and recalcitrant pool (RP-C) was not informed yet. I assume they used same tool as DOM.
4. Discussion section need some additional information and adjustments
In sub section 4.1, Because this paper emphasizes on Ultisol, known as the highly weathered soil in the title, I recommend author to provide concise explanation regarding the climate, particularly high precipitation (causing DOM leaching, particularly in more disturbed land use) at study area together with Ultisol properties (low C and N) in influencing the result of organic matter content obtained.
In sub section 4.1., the discussion needs to provide the explanation regarding the impact on land use management. As previously described in methods section, Bamboo plantation considered as the most frequently disturbed plantation, so this is also one of the main reasons why It has the lowest organic matter content, especially the light fraction content. Also mulch of regular pruning activity, contribute to high organic matter in tea plantation.
Minor comments
Line 42: word “ascribe” should be in past participle “ascribed”
Line 47-50: for clarity, I suggest change the sentence in to this “Zheng et al. (2017) observed that the high soil organic matter mineralization rate in the croplands, higher than the poplar plantations, was positively related with the bacterial abundance and the relative abundances of Actinobacteria and Bacteroidetes, while negatively correlated to the fungal abundance”
Line 54-70: I think it would be better to divide it in to two paragraphs, so easy to follow
Line 59: too wordy, I suggest change “when it with “ with “at”
Line 64: delete “, and” and put “while”
Line 69: I suggest, in one sentence, The author should indicate the importance of OM mineralization related the OM composition and microbial community in Ultisol
Line 82: Change “The climate is subtropical climate” in to “Study area is categorized as subtropical climate”
Line 81-99: Paragraph in subsection 2.1. would be better to separate in to two paragraphs
Line 95: I think you mean “an iced cooler box”
Line 96: “to prepare into “
Line 226: Insert “that” after revealed
Line 222-243: Too long paragraph, please split it in to 2 or 3 paragraphs
Line 260-275: I suggest splitting in to two paragraphs
Line 262 and 264: word “explained” was used so many times. Please consider using synonym i.e. “defined, clarified, etc”
Line 293-314: Please divide in to two or three paragraphs
Line 316-348: This is too long paragraph, I suggest to separate divide it in to two or three paragraphs
Line 328, 332, and 340: change “are” with “were”
Line 353-373: Please divide in to two or three paragraphs
Line 356: change “while” with “whereas”
Line 363: Use past participle word for passive sentence. Change “broke” in to “broken”
Line 373: insert “greater” after the
Line 379 and 381: Please consider using synonym for word “showed” like revealed, found out, etc
Author Response
General Comments
- The introduction should contain all the necessary information to justify the measurements. For instance, it should be introduced concisely about the Ultisol, in terms of organic matter contains and its related soil properties (such as N, C/N ratio).
Reply:
Thanks a lot for the comments, the suggestions of the reviewer were accepted. The description of “Ultisol widely distributes across subtropical and tropical zones of southern China. Ultisol suffers from strongly weathering and leaching, and has unique characteristics such as high contents of clay and Fe/Al and low contents of organic matter and nutrients.” was added in the lines of 63-65 revised manuscript.
- Sampling collection should explain in more detail information, i.e., i) tool used for soil sampling (like gauge auger), ii) the distance of soil sampling from the plant stand.
Reply:
Thanks a lot for the suggestions. The more details were supplemented in the lines of 91-93 revised manuscript. The description was that “The soils from ten randomly selected points (about 30 cm from the plant stand) were collected from 0-20 cm layer by 5 cm diameter auger, and mixed homogenously to form a composite sample in each plot.”.
- This part needs more detailed information.
In the methods section of 2.4 sub section, the method of fractionation did not apply/ or less information of substantial part as the reference paper (Rovira and Vallejo, 2022), like process of centrifugation, shaking the solution and drying the residue. Also, the tool used to analyse C concentration of labile C (LP I C, LP II C) and recalcitrant pool (RP-C) was not informed yet. I assume they used same tool as DOM.
Reply:
Thanks a lot for the comments, the suggestions of the reviewer were accepted. The detailed information weas supplemented in the lines 125-129 of revised manuscript. The description was that “The above hydrolysates were obtained by centrifuging at 10000 rpm for 30 min and then filtering through a 0.45-μm membrane filter. The organic carbon in the hydrolysates was determined by a TOC analyser (Vario TOC, Elementar, Germany). The organic carbon in the last residue was determined as the recalcitrant pool carbon (RP-C), calculated as the difference between the total organic carbon and the labile pool car-bon.”.
- Discussion section need some additional information and adjustments.
In sub section 4.1, Because this paper emphasizes on Ultisol, known as the highly weathered soil in the title, I recommend author to provide concise explanation regarding the climate, particularly high precipitation (causing DOM leaching, particularly in more disturbed land use) at study area together with Ultisol properties (low C and N) in influencing the result of organic matter content obtained.
In sub section 4.1., the discussion needs to provide the explanation regarding the impact on land use management. As previously described in methods section, Bamboo plantation considered as the most frequently disturbed plantation, so this is also one of the main reasons why It has the lowest organic matter content, especially the light fraction content. Also mulch of regular pruning activity, contribute to high organic matter in tea plantation.
Reply:
Thanks a lot for the suggestions. The discussion was adjusted and improved in the lines of 316-319 and 323-325 revised manuscript. The descriptions were that “Moreover, the mulch of regular pruning in the tea plantation could contribute to the high soil organic matter. Ultisol belongs to strongly weathered soil and had a limited capacity for organic matter accumulation, especially in the more disturbed bamboo plantation.” and “High precipitation in this area could stimulate DOM leaching, particularly in the more disturbed bamboo plantation.”.
Minor comments
Line 42: word “ascribe” should be in past participle “ascribed”.
Reply:
Thanks a lot for the suggest. The word “ascribe” was changed into “ascribed” in the line 40 of revised manuscript.
Line 47-50: for clarity, I suggest change the sentence in to this “Zheng et al. (2017) observed that the soil organic matter mineralization rate in the croplands, higher than the poplar plantations, was positively related with the bacterial abundance and the relative abundances of Actinobacteria and Bacteroidetes, while negatively correlated to the fungal abundance”.
Reply:
Thanks for the comments. The sentence was rewritten as “Zheng et al. (2017) observed that the soil organic matter mineralization rate in the croplands, higher than the poplar plantations, was positively related with the bacterial abundance and the relative abundances of Actinobacteria and Bacteroidetes, while negatively correlated to the fungal abundance.” in the lines 45-48 of revised manuscript.
Line 54-70: I think it would be better to divide it in to two paragraphs, so easy to follow.
Reply:
Thanks a lot for the suggest. The paragraph was divided into two paragraphs in the revised manuscript.
Line 59: too wordy, I suggest change “when it with” with “at”.
Reply:
Thanks a lot for the suggest. The phrase “when it with” was changed into “at” in the line of 55 of revised manuscript.
Line 64: delete “, and” and put “while”.
Reply:
Thanks a lot for the comments. The phrase “, and” was changed into “while” in the line 60 of manuscript.
Line 69: I suggest, in one sentence, The author should indicate the importance of OM mineralization related the OM composition and microbial community in Ultisol.
Reply:
Thanks a lot for the comments, the suggestions of the reviewer were accepted. The sentence was changed into “Therefore, it is very important to clearly understand the roles of both organic carbon components and microbial community compositions in the organic matter mineralization of Ultisol.” in the lines 66-68 of manuscript.
Line 82: Change “The climate is subtropical climate” in to “Study area is categorized as subtropical climate”.
Reply:
Thanks a lot for the comments. The sentence “The climate is subtropical climate” was changed into “Study area is categorized as subtropical climate” in the line 80 of manuscript.
Line 81-99: Paragraph in subsection 2.1. would be better to separate in to two paragraphs.
Reply:
Thanks a lot for the suggest. The paragraph was divided into two paragraphs in the revised manuscript.
Line 95: I think you mean “an iced cooler box”.
Reply:
Thanks a lot for the comments. The “an iced cooler” was changed into “an iced cooler box” in the line 94 of manuscript.
Line 96: “to prepare into”.
Reply:
Thanks a lot for the suggest. The phrase “to prepare as” was changed into “to prepare into” in the line 95 of manuscript.
Line 226: Insert “that” after revealed.
Reply:
Thanks a lot for the comments. The word “that” was inserted in the line 226 of manuscript.
Line 222-243: Too long paragraph, please split it in to 2 or 3 paragraphs.
Reply:
Thanks a lot for the suggest. The paragraph was divided into three paragraphs in the revised manuscript.
Line 260-275: I suggest splitting in to two or three paragraphs.
Reply:
Thanks a lot for the suggest. The paragraph was divided into three paragraphs in the revised manuscript.
Line 262 and 264: word “explained” was used so many times. Please consider using synonym i.e. “defined, clarified, etc”.
Reply:
Thanks a lot for the suggest. The word “explained” was changed into the synonym such as “interpret”, “define”, “clarify”, “identify”, and “account for” in the lines 271, 272, 275, and 277 of manuscript.
Line 293-314: Please divide in to two or three paragraphs.
Reply:
Thanks a lot for the suggest. The paragraph was divided into three paragraphs in the revised manuscript.
Line 316-348: This is too long paragraph, I suggest to separate divide it in to two or three paragraphs.
Reply:
Thanks a lot for the suggest. The paragraph was divided into two paragraphs in the revised manuscript.
Line 328, 332, and 340: change “are” with “were”.
Reply:
Thanks a lot for the suggest. The word “are” was changed into “were” in the lines 341, 345, and 351 of manuscript.
Line 353-373: Please divide in to two or three paragraphs.
Thanks a lot for the suggest. The paragraph was divided into two paragraphs in the revised manuscript.
Line 356: change “while” with “whereas”.
Reply:
Thanks a lot for the suggest. The word “while” was changed into “whereas” in the line 369 of manuscript.
Line 363: Use past participle word for passive sentence. Change “broke” in to “broken”.
Reply:
Thanks a lot for the suggest. The word “broke” was changed into “broken” in the line 375 of manuscript.
Line 373: insert “greater” after “the”.
Reply:
Thanks a lot for the comments. The word “greater” was inserted in the line 384 of manuscript.
Line 379 and 381: Please consider using synonym for word “showed” like revealed, found out, etc.
Reply:
Thanks a lot for the suggest. The word “showed” was changed into “exhibited” and “displayed” in the lines 391 and 393 of manuscript, respectively.
Reviewer 2 Report
Its a very interesting work that needs small improvments. My suggestions are presented in manuscript.

Author Response
Line 90 It would be interesting to know the composition of the fertilizer.
Reply:
Thanks a lot for the comments. The compound fertilizer contains N-P2O5-K2O at 20%-8%-8%. The description “The tea plantation was applied to 1500 kg ha-1 compound fertilizer (N-P2O5-K2O: 20%-8%-8%)” was in the line 88 of revised manuscript.
Line 90 What is the area of each plot?
Reply:
Thanks a lot for the suggestions. The area of each plot is 20 m × 20 m. The description “Three independent plots (20 m × 20 m) of each land-use type were randomly selected for soil sampling.” was in the line 90 of revised manuscript.
Line 102 It would be interesting to present a brief description of the method.
Reply:
Thanks a lot for the suggestions. In brief, the content of TOC was measured by potassium dichromate oxidation followed by titration with ferrous sulfate. Moreover, the content of TN was determined by the continuous steps of sulfuric acid digestion, sodium hydroxide distillation, boric acid uptake, and diluted acid titration. The description “In brief, the content of TOC was measured by potassium dichromate oxidation followed by titration with ferrous sulfate. Moreover, the content of TN was determined by the continuous steps of sulfuric acid digestion, sodium hydroxide distillation, boric acid uptake, and diluted acid titration.” was in the lines 102-105 of revised manuscript.
Line 107 During these days when were samples taken?
Reply:
Thanks a lot for the comments. The CO2 was collected on 1, 2, 4, 7, 11, 16, 22, 29, 37, 46, 56, 86, 101, and 121 days of the incubation period. The description “The CO2 released from soils was collected by syringes from the headspace of flasks and was measured on 1, 2, 4, 7, 11, 16, 22, 29, 37, 46, 56, 86, 101, and 121 days by gas chromatograph (Agilent, G7890A, Wilmington, DE, USA).” was in the lines 114-117 of revised manuscript.
Line 111 How is it collected? You must provide a more complete description.
Reply:
Thanks a lot for the comments. The caps of flasks were closed and the CO2 was collected from the headspace of flasks by syringes. The description “The caps of flasks were opened to replenish oxygen with the exception of regular CO2 collection. The CO2 released from soils was collected by syringes from the headspace of flasks and was measured on 1, 2, 4, 7, 11, 16, 22, 29, 37, 46, 56, 86, 101, and 121 days by gas chromatograph (Agilent, G7890A, Wilmington, DE, USA).” was in the lines 113-117 of revised manuscript.
Line 115 The word “aomunt” should be changed into “amount”.
Reply:
Thanks a lot for the suggestions. The word was changed into “amount” in the line 122 of revised manuscript.
Line 131 “All of the DOMs were adjusted to the same carbon concentration before the spectroscopic measurement.” How?
Reply:
Thanks a lot for the comments. In order to normalize fluorescence intensity (FI), all of the DOMs were adjusted by deionized water to the same carbon concentration before the spectroscopic measurement. The description “In order to normalize fluorescence intensity (FI), all of the DOMs were adjusted by deionized water to the same carbon concentration before the spectroscopic measurement.” was in the lines 140-142 of revised manuscript.
Line 184 It is necessary to improve the methodology, because in the description it seems that in these 125 days nothing has been done.
Reply:
Thanks a lot for the comments. The methodology of section 2.3 was improved in the revised manuscript. The description is that “The caps of flasks were opened to replenish oxygen with the exception of regular CO2 collection. The CO2 released from soils was collected by syringes from the headspace of flasks and was measured on 1, 2, 4, 7, 11, 16, 22, 29, 37, 46, 56, 86, 101, and 121 days by a gas chromatograph (Agilent, G7890A, Wilmington, DE, USA).”.
Line 292 I would like to see a better explanation of the reason for the variation between treatments.
Reply:
Thanks a lot for the comments. The section 4.1 was fully improved in the revised manuscript. The descripts were that “Moreover, the mulch of regular pruning in the tea plantation could contribute to the high soil organic matter. Ultisol belongs to strongly weathered soil and had a limited capacity for organic matter accumulation, especially in the more disturbed bamboo plantation.” in lines 323-326, “The Cam and the Tea showed more contents of DOM in soils than that in the Bam (Table 3), the variation was similar to the contents of LP I-C and LP II-C among the land-use types.” in lines 327-329, “High precipitation in this area could stimulate DOM leaching, particularly in the more disturbed bamboo plantation.” in lines 332-333, and “The Cam and the Tea provided more plant materials, which would leach abundant DOMs with simple structure compared with the Bam, because the DOM from plant materials mostly showed low molecular complexity and humification indices. Furthermore, the higher activities of enzymes (Table 4) perhaps resulted in more input of DOMs with simple structure in the Cam and the Tea than that in the Bam.” in lines 334-339.